# Utilizing Pruning and Leaf Removal to Optimize Ripening of *Vitis riparia*-Based 'Frontenac Gris' and 'Marquette' Wine Grapes in the Northern Great Plains

**Andrew Aipperspach [1,†], James Hammond [2] and Harlene Hatterman-Valenti [2,*]**

[1] Former Graduate Student in Plant Sciences, North Dakota State University, Fargo, ND 58105, USA; andrew.aipperspach@gmail.com

[2] Department of Plant Sciences, North Dakota State University, Fargo, ND 58105, USA; james.hammond@ndsu.edu

[*] Correspondence: h.hatterman.valenti@ndsu.edu

[†] Current address: SESVanderHave, 5908 52 Ave S., Fargo, ND 58104, USA.

**Abstract:** Experiments were conducted to evaluate the effects of three pruning levels (20, 30 and 40 nodes per vine) and three fruit-zone leaf removal levels (0%, 50%, and 100%) on the yield and fruit quality of Frontenac gris and Marquette wine grapes in a northern production region. The study was conducted at three North Dakota vineyards located near Buffalo, Clifford, and Wahpeton, North Dakota, in 2011 and 2012. Increasing the number of buds retained increased yields and reduced pruning weights in both cultivars. Frontenac gris and Marquette yields were greatest when vines had 50% of the fruit-zone leaves removed due to heavier clusters, suggesting that the 100% fruit-zone leaf removal level was too severe. Individual berries in clusters were also heavier when vines were pruned to retain 40 buds. Frontenac gris fruit quality was similar both years and was not influenced by pruning or leaf removal levels. Marquette fruit total soluble solids content was greater in 2012 due to the warmer and longer growing season. Marquette fruit titratable acidity was lower when 100% of the fruit-zone leaves were removed. These results suggest that for the two cold-hardy hybrid wine grapes used in this study, greater bud retention levels should be investigated. Results also warrant further research into cultivar adaptiveness to northern Great Plains conditions. With further research, it is anticipated that wine grape cultivars and management practices will be identified to produce acceptable yields and fruit quality for commercial wine grape production.

**Keywords:** cold-hardy hybrid grapes; cluster exposure; canopy management

## 1. Introduction

The development of cold, hardy, *Vitis riparia*-based grape cultivars started a new industry in the northern Great Plains region. In North Dakota, the passing of farm winery legislation in 2001 signified the start of the grape and wine industry [1]. These *V. riparia*-based grape cultivars have enabled wine grape production in areas where *V. vinifera* cultivars cannot survive winter low temperatures, but they also present new challenges as fruit development and fruit composition differs from *V. vinifera* cultivars [2–4]. One of the greatest challenges with *V. riparia*-based cultivars in the northern Great Plains region is the high titratable acidity (TA) levels. Research is limited on viticulture methods to reduce high TA levels, especially under the short growing season and low accumulated heat units in the northern Great Plains region [4].

Pruning is a critical part of grape production to control crop load and vine growth in order to manipulate crop yield and fruit quality [5–7]. The influence of crop level on fruit quality with *V. vinifera* cultivars is inconclusive, with similar reports showing a low crop load associated with high wine



quality has no relationship between crop level and wine quality [8–10]. Studies have determined the effect of crop level on specific cold-hardy grape cultivars, but these studies were conducted around the Great Lakes area, with longer growing seasons and milder winter conditions [11,12]. While the need for adequate vegetative growth has been recognized as it relates to sufficient carbohydrate production and in turn, full fruit ripening, the negative impact of over-cropping from delayed fruit maturity, compromised quality, and reduced winter hardiness can be exacerbated in cool climates [13,14]. Carbohydrates are necessary for shoot lignification and energy storage for overwintering, insufficient storage amounts detrimentally affect grapevine growth and overall health [15]. Balanced pruning methods allow growers to better anticipate and control reproductive yield and achieve a sustainable balance of plant growth and yield within their vineyard.

In cold climate regions, limited accumulation of heat units during the growing season often leads to insufficient grape fruit ripening [4]. Fruit-zone leaf removal has been shown to accelerate the soluble solids and flavonoid accumulation in *V. vinifera* fruit by increasing the rate of berry ripening due to control of microclimate in warmer regions [10,16–18]. Increased rate of berry ripening may shorten the time to harvest and allow increased time for post-harvest carbohydrate accumulation within the vine prior to dormancy [11]. Exposing fruit to high solar radiation, which increases the rate of grape berry ripening also has the potential to reduce disease incidence and increase grape quality at harvest in northern production regions [4,13,19]. Adequate light to grape vine buds during floral development and differentiation is also critical for subsequent bud fertility and fruitfulness [20,21]. In cold climate conditions, the ability of a grapevine to overwinter is critical for consistent annual production. The objective of this study was to evaluate pruning levels and fruit-zone leaf removal levels for consistent yield and quality of Frontenac gris and Marquette in North Dakota.

## 2. Materials and Methods

### 2.1. Vineyard Sites and Cultural Practices

Experiments were conducted at three vineyards in 2011 and 2012, near Buffalo, Clifford, and Wahpeton, ND. All three vineyards were within the Red River Valley in southeast to northeast North Dakota. Two cultivars, Marquette and Frontenac gris, were selected for the study since they were the main red and white wine grape cultivar used for production in North Dakota, respectively. Both cultivars are University of Minnesota introductions, and have been planted rather extensively in North Dakota and other northern climates for the last 10 years [22].

The soil within the experimental portion of the Buffalo vineyard was a Barnes-Buse loam classified as a fine-loamy, mixed, 70 superactive, frigid Calcic Hapludoll with 3.6% OM and a pH of 7.7 [23]. The vineyard had 500 Marquette vines and 200 Frontenac gris vines. All vines were five years old, planted on their own roots in a north/south direction with 2.4 m between vines and 3.0 m between rows, grass alleys, and trained to a mid-wire cordon trellis system with vertical shoot positioning. Within-row weed control consisted of an early spring application of glufosinate-ammonium (Rely 280®, BASF Corp., Research Triangle Park, Durham, NC, USA) and flumioxazin (Chateau®, Valent USA. Corporation, Walnut Creek, CA, USA). No fungicides, supplemental water, or fertilizer were used. Dormant pruning to 2-bud spurs occurred on 27 April 2011 and 10 May 2012.

The soil at the Clifford vineyard was a Gardena silt loam classified as a coarse-silty, mixed, superactive, frigid Pachic Hapludoll, with 4.3% OM and a pH of 7.7. The vineyard had 300 Marquette vines and 200 Frontenac gris vines. All vines were four years old, planted on their own roots in a north/south direction with 2.4 m between vines and rows, grass alleys, and trained to a mid-wire cordon trellis system with vertical shoot positioning. Within-row weed control consisted of an early spring application of glufosinate-ammonium and flumioxazin. No fungicides, supplemental water, or fertilizer were used. Dormant pruning to 2-bud spurs occurred on 25 April 2011 and 5 May 2012.

The soil at the Wahpeton vineyard was an Aberdeen-Ryan silty clay loam classified as a fine, smectitic, frigid Glossic Natrudoll with 3.7% OM and a pH of 7.4. The vineyard had 250 Marquette vines

and 250 Frontenac gris vines. All vines were five years old, planted on their own roots in a north/south direction with 2.4 m between vines and rows, grass alleys, and trained to a mid-wire bilateral cordon trellis system with vertical shoot positioning. The vineyard was not irrigated, no fungicides were applied, and only minor fertilization (<5.6 kg ha$^{-1}$ N) took place in the spring of 2012. Weed control was carried out with a home-made tillage implement that was used only during early spring and supplemented with hand weed removal. Dormant pruning to 2-bud spurs occurred on 28 April 2011 and 4 May 2012.

Weather data was collected for each location from North Dakota Agriculture Weather Network (NDAWN) weather stations that were located nearest to the vineyard [24]. Each weather station was less than 20 km from a vineyard. Growing Degree Days were obtained from NDAWN using the calculations:

Daily average temperature (°C) = (daily max. temp. + daily min. temp.)/2
Daily grape GDD (°C) = daily avg. temp. −10°

## 2.2. Treatments and Vine Parameters

Three primary bud retention treatments were implemented to maintain 20, 30 or 40 primary buds per vine. Dormant pruning weights were taken as pruning weights are generally well correlated with other components of vine size. The pruning of dormant canes was completed after measuring the longest one-year-old cane on each cordon to examine vine vigor. After pruning to the experimentally defined number of primary buds, the weight of pruned wood for each plant was recorded using a digital scale (A&D SK-1000, A&D Engineering, Inc., San Jose, CA, USA). Following bud break and shoot elongation of 5 to 10 cm, shoot thinning was conducted to remove non-count buds that had broken and other growth not congruent with treatment parameters. This standard of pruning was continued throughout the season to remove any growth that would arise from non-count buds and suckers.

Fruit-zone leaf removal treatments were implemented as zero percent (control), 50 or 100 percent to increase cluster exposure to sunlight. In the 100 percent exposure treatments, leaves were removed from primary shoot nodes one through six, while in the 50 percent exposure treatments, leaves were removed from primary shoot nodes one through six, but only the east-facing side. Leaf removal occurred after fruit set, at modified Eichhorn–Lorenz growth stage 29 (grape size ranged from $\frac{1}{2}$ cm to $\frac{3}{4}$ cm) [25]. In 2011, fruit-zone leaf removal occurred approximately on 25 July, while in 2012 this occurred approximately on 12 July for the three locations. Primary shoots averaged two clusters for each cultivar.

The progressive ripening of fruit was evaluated weekly, starting 21 d post-veraison by measuring soluble solid concentration (SSC) and pH. Eight berries were taken from the cluster center of eight randomly selected clusters on each vine pair weekly and analyzed for pH (Mettler Toledo S20 pH meter, Mettler-Toledo Inc., Columbus, OH, USA) and SSC (Extech Portable Refractometer, Extech Instruments, Nashua, NH, USA). All sample berries were obtained from the middle of their respective clusters, in order to maintain consistency when sampling and avoid variable ripeness between the upper berries in a cluster and lower berries of the same cluster.

Time of final berry harvest was dictated by environmental conditions such as a killing frost, which signaled the end of the growing season; grape ripeness with acceptable pH and SSC measurements [26]; and economic considerations, such as a buyer's request. Grape yields were obtained by harvesting each grape vine individually and measured by the total weight of grapes harvested per vine and then converted to a per area basis for the specific row and vine spacing. During harvest, the number of grape clusters per vine was counted, and the weight of all grapes from each vine was recorded in the field. A random two-cluster sample was collected from each vine for further analysis. Collected clusters were stored at 7 °C until analysis within two days of harvest. For the final berry composition data, a randomly selected 50 berry sub-sample from the collected cluster samples were juiced in their entirety for soluble solids concentration and pH as previously described, as well as titratable

acidity (TA) determination using a mini-titrator (Hannah Instruments HI 84102 Mini-titrator Hannah Instruments, Smithfield, RI, USA).

## 2.3. Statistical Analysis

The experimental design at each vineyard was a randomized complete block with three factors (cultivars, retained primary buds, and leaf removal percentage) and four replications. Two consecutive vines were used for each experimental unit, except when vine growth/vigor appeared to be limited and inconsistent. This only occurred at the Wahpeton vineyard, and in such cases, the smaller vine was skipped and substituted with the next vine in the row. Location (vineyards) and year were considered random effects, while retained primary buds, fruit-zone leaf removal percentage, and cultivar were considered fixed effects. All data except berry sampling prior to harvest were analyzed using the General Linear Models Procedure (SAS version 9.3, Statistical Analysis Systems Institute, Cary, NC, USA). Data from three vineyards were combined after conforming to the test of homogeneity of variance developed by Brown and Forsythe [27]. Treatment means were separated where appropriate using Duncan's Means Separation test at the 0.05 level of significance. Statistical analysis was not performed on weekly fruit sampling since the intent was to collect enough fruit to get an estimate of fruit SSC and pH change over time.

## 3. Results

### 3.1. Phenology

Late winter and early spring environmental conditions (February–May) were similar for the three locations, even though there was more than 200 km latitudinal separation between vineyards (Table 1). However, the average monthly temperatures in 2012 were considerably milder than 2011 or the 30-year average. Above average winter and spring temperatures in 2012 encouraged early bud break and the accumulation of more growing degree days (GDD) using a base temperature of 10 °C (Table 2). In an effort to discourage bud break and avoid potential spring frost injury in 2012, dormant cane pruning was delayed approximately 10 d. Spring frost injury did not occur and above average spring temperatures resulted in an accelerated rate of growth and development in 2012 when phenology dates were compared to 2011 (data not shown), regardless of the rainfall differences over the same period (Table 3). Fruit harvested in 2012 received more accumulated GDDs than the recommended 1400 GDD °C to ripen the fruit for the grape cultivars released by the University of Minnesota [28].

**Table 1.** Average monthly air temperatures for 2011 and 2012, and normal monthly air temperatures (°C) at field locations collected from nearest North Dakota Agriculture Weather Network (NDAWN) weather station (in parenthesis).

| Month | Buffalo (Prosper) | | | Clifford (Galesburg) | | | Wahpeton | | |
|---|---|---|---|---|---|---|---|---|---|
| | 2011 | 2012 | 30-Year Average | 2011 | 2012 | 30-Year Average | 2011 | 2012 | 30-Year Average |
| | | | | | °C | | | | |
| January | −17 | −8 | −13 | −17 | −8 | −14 | −17 | −7 | −12 |
| February | −13 | −6 | −10 | −12 | −7 | −11 | −13 | −6 | −9 |
| March | −8 | 4 | −3 | −8 | 3 | −4 | −6 | 5 | −2 |
| April | 5 | 8 | 6 | 5 | 8 | 6 | 6 | 9 | 7 |
| May | 12 | 15 | 13 | 11 | 15 | 13 | 12 | 16 | 15 |
| June | 19 | 20 | 19 | 18 | 20 | 18 | 19 | 20 | 20 |
| July | 23 | 24 | 21 | 23 | 23 | 21 | 23 | 24 | 22 |
| August | 21 | 20 | 20 | 21 | 20 | 20 | 20 | 19 | 18 |
| September | 15 | 15 | 15 | 15 | 14 | 15 | 15 | 15 | 16 |
| October | 11 | 6 | 7 | 10 | 5 | 6 | 11 | 6 | 8 |
| November | 0 | −2 | −2 | 0 | −3 | −3 | 0 | −1 | −1 |
| December | −5 | −10 | −10 | −5 | −11 | −12 | −4 | −10 | −9 |

**Table 2.** Accumulated growing degree days data (base 10 °C) during 2011 and 2012 for 1 April to 30 September growing seasons collected from NDAWN weather stations' (in parenthesis) nearest vineyard locations and the comparison to the 5-year average.

| | Buffalo (Prosper) | | Clifford (Galesburg) | | Wahpeton | |
|---|---|---|---|---|---|---|
| | **2011** | **2012** | **2011** | **2012** | **2011** | **2012** |
| AGDD$^Z$ (10 °C) | 1371 | 1594 | 1314 | 1502 | 1357 | 1617 |
| Departure from 5-year average | +39 | +269 | +11 | +213 | +55 | +223 |

$^Z$ Abbreviation AGDD = Accumulated growing degree days.

**Table 3.** Monthly rainfall totals from 2011 and 2012 measured for Clifford, Wahpeton, and Buffalo locations, collected from the nearest NDAWN weather station (in parenthesis).

| | Clifford (Galesburg) | | Wahpeton | | Buffalo (Prosper) | |
|---|---|---|---|---|---|---|
| **Month** | **2011** | **2012** | **2011** | **2012** | **2011** | **2012** |
| | | | mm | | | |
| April | 35 | 36 | 36 | 81 | 45 | 46 |
| May | 101 | 26 | 69 | 37 | 81 | 46 |
| June | 102 | 16 | 87 | 75 | 132 | 67 |
| July | 102 | 7 | 146 | 46 | 151 | 16 |
| August | 89 | 11 | 122 | 58 | 89 | 23 |
| September | 13 | 3 | 13 | 9 | 6 | 15 |
| Total | 442 | 99 | 473 | 306 | 504 | 213 |

The longest one-year-old canes were influenced by the interaction of cultivars and number of retained primary buds. For both cultivars, the longest one-year-old canes occurred when vines were pruned to retain 40 primary buds, while the shortest one-year-old canes occurred when vines were pruned to retain 30 primary buds (Table 4). However, Frontenac gris, vines pruned to retain 20 primary buds, had similar length for one-year-old canes as vines pruned to retain 40 primary buds. Further analysis suggested that the interaction was due to a degree of magnitude from genetic differences. Marquette produced longer one-year-old canes and was considered more vigorous compared to Frontenac gris.

**Table 4.** Influence of bud retention on length of longest one-year-old cane and weight of pruned dormant canes for two *V. riparia*-based cultivars averaged over fruit-zone leaf removal levels, locations, and years.

| Buds Retained | Frontenac Gris | | Marquette | |
|---|---|---|---|---|
| | **cm** | **g** | **cm** | **g** |
| 20 | 209.8 a $^z$ | 365.1 a | 241.8 b | 1031.9 a |
| 30 | 183.9 b | 414.2 a | 227.4 c | 883.5 b |
| 40 | 205.0 a | 247.4 b | 283.7 a | 925.1 b |

$^z$ Means in a column followed by the same letter are not significantly different according to Duncan's multiple range test at $p < 0.05$.

Dormant cane pruning weights differed between years and from a cultivar by primary bud retention level interaction. Dormant cane pruning weights for Frontenac gris and Marquette were approximately 75% and 72% greater in 2011 compared to 2012, respectively. This was attributed to the increased training and pruning in 2011 compared to grower practices the previous year. Frontenac gris vines pruned to retain 20 and 30 buds had greater one-year-old cane pruning weights, approximately a 32% and 40% increase, respectively, compared to vines that had 40 buds retained (Table 4). Marquette

vines pruned to retain 20 buds had greater one-year-old wood pruning weights, approximately 25% and 11% more than vines that had 30 and 40 buds retained, respectively.

## 3.2. Fruit Yield

Grape yield was influenced by the three-way interaction between cultivar, bud retention, and fruit-zone leaf removal percentages, and the two-way interaction between cultivar and fruit-zone leaf removal percentages. Fruit yield increased as the number of buds retained increased for both cultivars, with the exception of Frontenac gris, with 50% of the fruit-zone leaves removed (Figure 1). The highest yield with Frontenac gris occurred when 40 buds were retained and no fruit-zone leaves were removed, while the lowest yield occurred when 20 buds were retained and 100% of the fruit-zone leaves were removed. Results suggest that the severity of leaf removal for the less vigorous Frontenac gris at the Eichhorn–Lorenz growth stage 29 with only 20 shoots per vine may have further reduced the yield potential, due to insufficient leaf area from the leaf removal and shoot thinning of non-count buds and suckers. Marquette yield was lowest when 20 buds were retained and no fruit-zone leaves were removed. However, yields were similar when 20 buds were retained for all three fruit-zone leaf removal treatments. The highest yield for Marquette occurred when 40 buds were retained and 50% of the fruit-zone leaves were removed. However, yields were again similar when 40 buds were retained for all three fruit-zone leaf removal treatments.

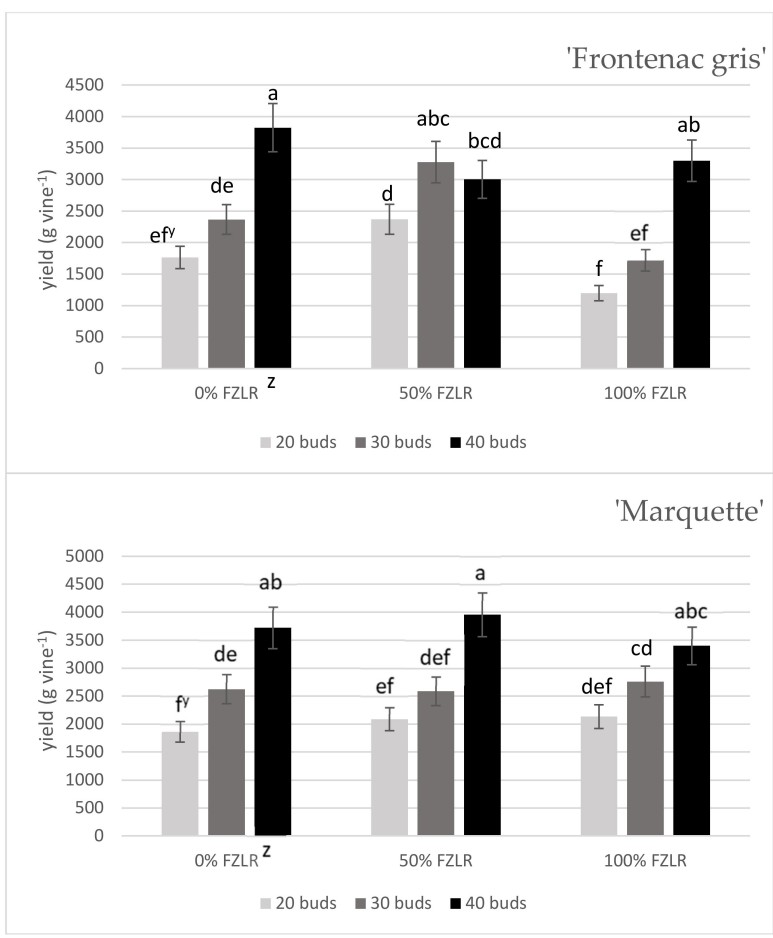

**Figure 1.** Effect of grape cultivar, bud retention, and fruit-zone leaf removal percentage on fruit yield of two *V. riparia*-based cultivars averaged over locations and years; [z] abbreviation FZLR = fruit-zone leaf removal; [y] means followed by the same letter(s) are not significantly different according to Duncan's multiple range test at $p < 0.05$.

　　　An analysis of the two-way interaction between cultivar and fruit-zone leaf removal percentages showed that the greatest yield for both cultivars was when 50% of the fruit-zone leaves were removed (Table 5). The lowest yield with Frontenac gris occurred when vines had 100% of the fruit-zone leaves removed. The lowest yield with Marquette occurred when vines either had no leaf removal or 100% fruit-zone leaf removal. Differences between cultivars was attributed to the lower vigor with Frontenac gris vines.

**Table 5.** Effect of fruit-zone leaf removal level on fruit yield for two *V. riparia*-based cultivars averaged over primary bud retention levels, locations, and years.

| Leaf Removal Percentage | Frontenac Gris | | Marquette | |
|:---:|:---:|:---:|:---:|:---:|
| | kg ha$^{-1}$ | | | |
| 0 | 1803 | b $^z$ | 1862 | b |
| 50 | 1961 | a | 1957 | a |
| 100 | 1410 | c | 1880 | b |

$^z$ Means in a column followed by the same letter are not significantly diffe rent according to Duncan's multiple range test at $p < 0.05$.

　　　Average fruit cluster weight derived from the total vine yield and the number of clusters per vine at harvest were used to help explain vine and vineyard productivity. The average fruit cluster weight was influenced by the three-way interaction between cultivar, bud retention, and fruit-zone leaf removal percentages and the two-way interaction between cultivar and fruit-zone leaf removal percentages. Frontenac gris vines with 50% fruit-zone leaves removed and 30 buds retained had the heaviest fruit clusters (Figure 2). However, these clusters were only heavier than fruit clusters from vines with no leaves removed and 20 buds retained. Marquette vines with 100% fruit-zone leaves removed and 40 buds retained had the heaviest fruit clusters. These fruit clusters were heavier than fruit clusters from all other fruit-zone leaf removal and retained bud combination treatments, except when vines had no leaves removed and 30 buds retained. When analyzing the two-way interaction between cultivar and fruit-zone leaf removal percentages, Frontenac gris vines with 50% fruit-zone leaf removal had the heaviest fruit clusters when compared to vines with 100% fruit-zone leaves removed (Table 6). In contrast, Marquette vines with 100% fruit-zone leaves removed had heavier fruit clusters when compared to vines with 50% fruit-zone leaves removed.

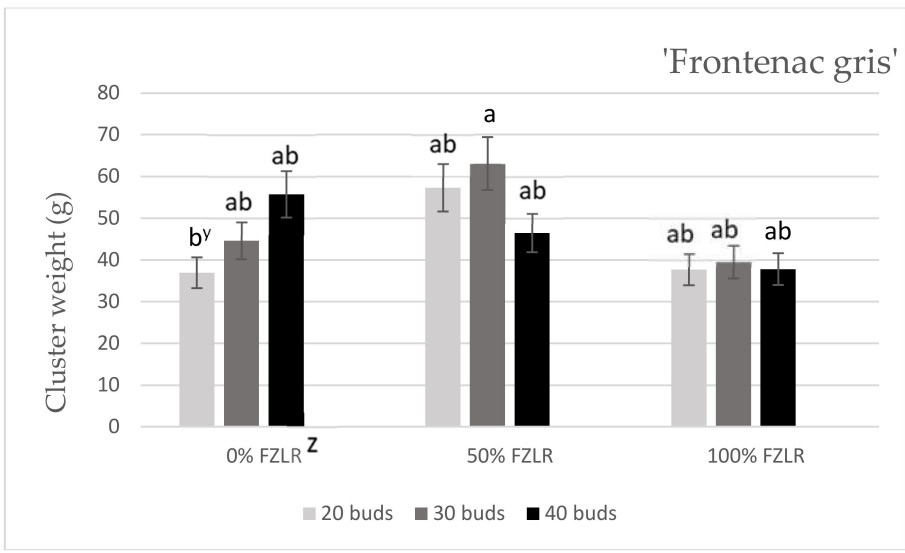

**Figure 2.** *Cont*.

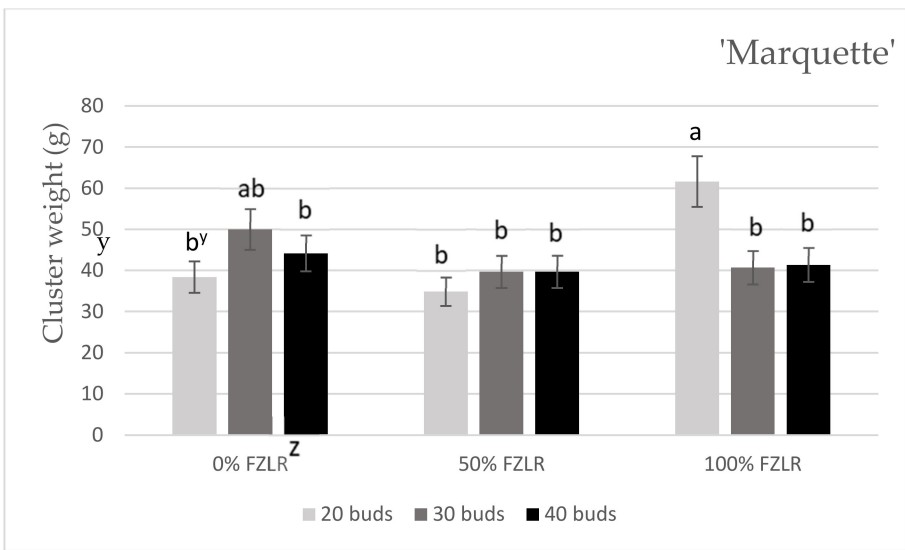

**Figure 2.** Effect of grape cultivar, bud retention, and fruit-zone leaf removal percentage on cluster weight of two *V. riparia*-based cultivars averaged over locations and years; [z] Abbreviation FZLR = fruit-zone leaf removal; [y] Means followed by the same letter(s) are not significantly different according to Duncan's multiple range test at $p < 0.05$.

**Table 6.** Effect of fruit-zone leaf removal on cluster weight at harvest for two *V. riparia*-based cultivars averaged over primary bud retention levels, locations, and years.

| Leaf Removal Percentage | Frontenac Gris | | Marquette | |
|:---:|:---:|:---:|:---:|:---:|
| | **g** | | | |
| 0 | 44.954 | ab [z] | 44.382 | ab |
| 50 | 46.907 | a | 39.345 | b |
| 100 | 37.900 | b | 49.541 | a |

[z] Means in a column followed by the same letter(s) are not significantly different according to Duncan's multiple range test at $p < 0.05$.

The interaction between cultivar and pruning level affected berry weight. Frontenac gris vines pruned to retain 20 buds had the lightest berry weight, while vines pruned to retain 40 buds had the heaviest berry weight (Table 7). However, berry weights were not statistically different. Marquette vines pruned to retain 30 buds had the lightest berry weight, while vines pruned to retain 40 buds had the heaviest berry weight. Again, berry weights were not statistically different, suggesting that berry weight differences were minimal and not the result of treatment effects.

**Table 7.** Influence of bud retention on berry weight at harvest for two *V. riparia*-based cultivars averaged over fruit-zone leaf removal levels, locations, and years.

| Buds retained | Frontenac Gris | | Marquette | |
|:---:|:---:|:---:|:---:|:---:|
| | **g** | | | |
| 20 | 0.92 | a [z] | 1.01 | a |
| 30 | 0.96 | a | 0.99 | a |
| 40 | 0.98 | a | 1.04 | a |

[z] Means in a column followed by the same letter are not significantly different according to Duncan's multiple range test at $p < 0.05$.

### 3.3. Fruit Composition

Fruit SSC and pH data were collected weekly to assist with the evaluation of fruit ripening starting 28 days after the start of veraison in 2011, and 21 days after the start of veraison in 2012, due to the perceived warmer growing season. Data were averaged over fruit-zone leaf removal percentages for the graphs showing fruit ripening, in response to the number of buds retained on a vine (Figure 3). The SSC for Frontenac gris and Marquette appeared to be similar up to 35 days after veraison, except when vines had 30 buds retained in 2012. However, at 42 and 48 days after veraison, the SSC for both cultivars was greater in 2012 than 2011. Frontenac gris fruit pH was rather similar for all sampling dates. Likewise, Marquette fruit pH was rather similar between years until 48 days after veraison, when pH values rose to approximately 3.4 in 2012, while the pH values in 2011 remained at approximately 3.1.

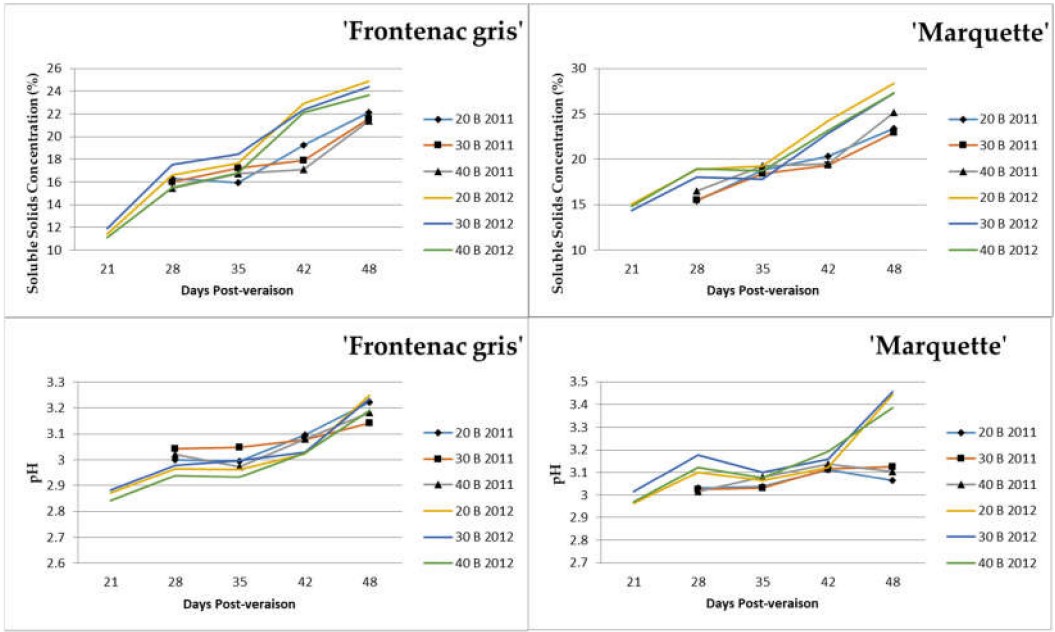

**Figure 3.** Weekly fruit sampling to determine soluble solid concentration and pH as parameters of fruit maturity for two *V. riparia*-based cultivars averaged over fruit-zone leaf removal levels and locations; [z] Abbreviation B = number of primary buds retained.

Data were averaged over the number of buds retained for the graphs showing fruit ripening in response to fruit-zone leaf removal percentages on a vine (Figure 4). The SSC for Frontenac gris appeared to be similar up to 35 days after veraison, except when vines had 50% fruit-zone leaves removed in 2012. However, at 42 and 48 days after veraison, the SSC for Frontenac gris was greater in 2012 than 2011. Fruit SSC for Marquette also showed greater separation between years when fruit was sampled 42 and 48 days after veraison. In addition, the SSC in 2011 at 48 days after veraison for fruit collected from vines with 100% fruit-zone leaf removal was lower than the SSC for fruit collected from vines with 0% or 50% fruit-zone leaf removal. Frontenac gris fruit pH was rather similar for all sampling dates. Marquette fruit pH was greater in 2012 compared to 2011, at 28 and 48 days after veraison. In addition, fruit pH in 2012 at 48 days after veraison for fruit collected from vines with 100% fruit-zone leaf removal was lower than fruit pH for fruit collected from vines with 0% or 50% fruit-zone leaf removal.

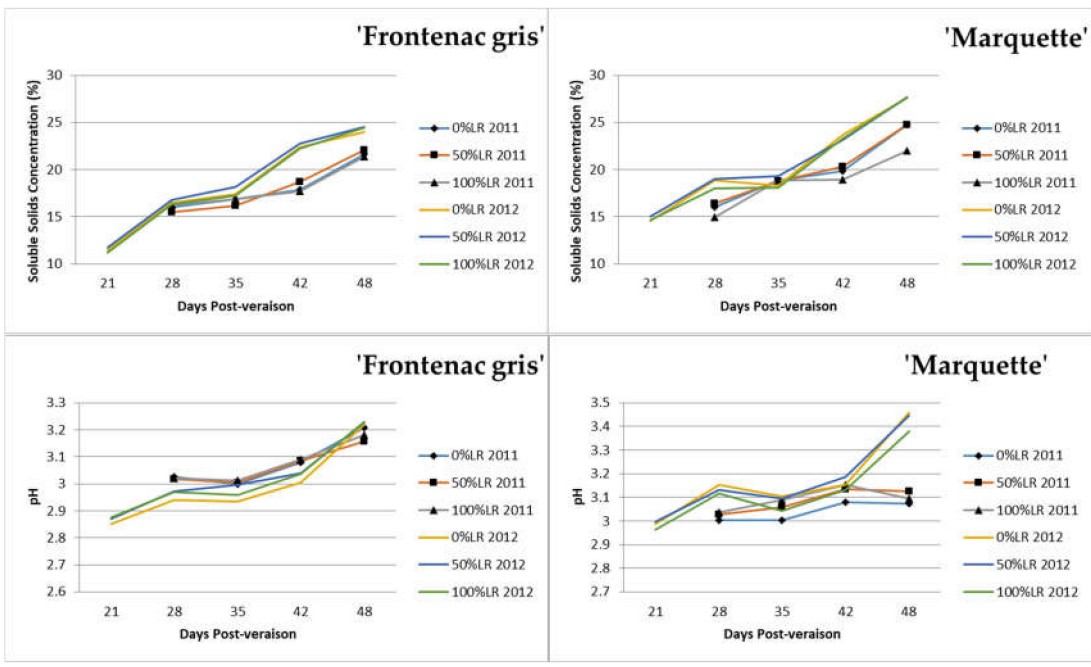

**Figure 4.** Weekly fruit sampling to determine soluble solid concentration and pH as parameters of fruit maturity for two *V. riparia*-based cultivars averaged over the number of retained primary buds and locations; [z] Abbreviation LR = percentage of fruit-zone leaves removed.

Even though bud retention directly influenced fruit yield, the SSC and pH fruit composition parameters at harvest were similar regardless of the pruning level. An interaction between year and cultivar affected SSC measured in Brix. Marquette fruit harvested in 2012 had higher SSC compared to grapes harvested from the same vines in 2011 (Table 8). However, Frontenac gris fruit had similar SSC percentages each year. Continued analysis of fruit pH levels revealed no differences from pruning or fruit-zone leaf removal levels, regardless of the cultivar (data not shown).

**Table 8.** Effect of year on soluble solid concentration at harvest for two *V. riparia*-based cultivars averaged over locations, retained primary buds, and fruit-zone leaf removal levels.

| Year | Frontenac Gris | | Marquette | |
|------|------|------|------|------|
| | | Brix | | |
| 2011 | 21.40 | a [z] | 22.95 | b |
| 2012 | 20.89 | a | 27.07 | a |

[z] Means in a column followed by the same letter(s) are not significantly different according to Duncan's multiple range test at $p < 0.05$.

Pruning level did not influence TA at harvest. However, the interaction between cultivars and fruit-zone leaf removal levels did influence TA. Marquette fruit from the 100% fruit-zone leaf removal treatment had lower TA levels compared to fruit from vines receiving 50% or 0% fruit-zone leaf removal (Table 9). Fruit from vines which received 100% fruit-zone leaf removal had a TA of 6.40 g $L^{-1}$, approximately 2 g $L^{-1}$ less than the fruit from vines with no fruit-zone leaf removal. In contrast, fruit-zone leaf removal did not affect TA levels in Frontenac gris.

**Table 9.** Effect of fruit-zone leaf removal on titratable acidity at harvest for two *V. riparia*-based cultivars averaged over pruning levels, locations, and years.

| Leaf Removal Percentage | Frontenac Gris | | Marquette | |
|:---:|:---:|:---:|:---:|:---:|
| | g/L | | | |
| 0 | 8.17 | a [z] | 8.19 | a |
| 50 | 8.02 | a | 7.82 | a |
| 100 | 8.40 | a | 6.40 | b |

[z] Means followed by the same letter(s) are not significantly different according to Duncan's multiple range test at $p < 0.05$.

## 4. Discussion

Average monthly air temperatures and rainfall along with accumulated growing degree day data for the three vineyards suggests greater variation between years than between locations. All three vineyards are located within the Red River Valley, one of three distinct geographical regions in North Dakota [29]. The Red River Valley soil was formed from the sediment that settled out of the ancient glacial Lake Agassiz and is considered one of the most fertile soils in the world and extremely productive. Yearly variation of environmental conditions was most evident from the milder winter conditions, combined with slightly warmer spring conditions in 2012, compared to 2011 and to the 30-year average. As a result, vine bud swell and emergence from winter dormancy was earlier in 2012 compared to 2011. Earlier bud break and slightly warmer June and July temperatures in 2012 also led to earlier flowering, earlier fruit-zone leaf removal, and earlier veraison for both cultivars. Hatterman-Valenti et al. [30] examined the growth and yield of 16 cultivars over a seven-year period and reported that cultivars showed no consistency for phenophases among years, indicating erratic spring climate conditions in the region. Similarly, Malheiro et al. [31] showed the effects of temperature on phenology, especially how an increase in temperature during the spring will advance the timing of all phenophases.

One-year-old cane lengths were taken in an attempt to help explain plant vigor along with dormant pruning weights in response to crop load (primary bud retention levels). Previous research has shown that grapevines have adaptive response processes following differential cropping, such as reduced budburst, reduced fruitfulness, shorter shoots with closer internodes, increased yield, and lighter clusters with smaller berries [32–34]. The current results were just the opposite, with the longest one-year-old canes from vines pruned to retain 40 buds, the highest crop load level. However, as Greven et al. [33] identified, most of these studies were conducted only 1–2 years and needed 4–7 years to show the sustainability of node pruning regimes and the potential change in vine behavior. Marquettes' consistently longer one-year-old canes at each crop load level averaged over three locations and two years compared to Frontenac gris suggests that Marquette is more vigorous than Frontenac gris under Red River Valley conditions.

Dormant pruning weights provide information on vine size and vigor when used with fruit production; they can help determine crop load to assist with the assessment of vine balance. The dormant pruning weight differences between years was expected primarily due to increased pruning and training in 2011, compared to pruning the previous year, and from the better growing conditions in 2010 compared to 2011. Grant et al. [35] showed that even with cold-hardy hybrid grape cultivars; shoot growth can vary greatly with temperature differences during the growing season. Dormant pruning weights were approximately two to four times greater at each primary bud retention level for Marquette vines compared to Frontenac gris, reinforcing the suggestion that Marquette is more vigorous than Frontenac gris. Atucha et al. [36] also noted the vigorous growth of Marquette in cool-climate conditions, particularly as lateral shoots, but did not have Frontenac gris in the cultivar comparison study.

Berries were sampled weekly to obtain fruit SSC and pH estimates toward the harvest indices winemakers desired. Data were averaged over locations, years, and the fruit-zone leaf removal

percentages and over locations, years, and the number of retained primary buds. Graphs demonstrated how warmer growing conditions (higher average monthly air temperatures and greater AGDD) in 2012 compared to 2011 can hasten fruit ripening, especially for Marquette. Berry sampling at 48 days after veraison also reinforced fruit SSC and pH results at harvest.

Fruit yield differences were attributed to the primary bud retention level maintained throughout the growing season and that increased production was possible through the retention of more buds. Grape vines pruned to retain 20 primary buds produced the least amount of fruit with an average of 1903 g vine$^{-1}$ or 1294 kg ha$^{-1}$. Retaining an additional 10 buds increased the yield approximately 34% to 1739 kg ha$^{-1}$. Grape vines pruned to retain 40 primary buds were the highest yielding, with an average of 3534 g vine$^{-1}$ or 2403 kg ha$^{-1}$, an 86% yield increase by retaining twice as many buds. The only exception was Frontenac gris vines with 50% of the fruit-zone leaves removed. Other researchers have also shown that an increase in crop load will result in an increase in crop yield [32–34]. However, Greven et al. [33] showed how a vine will moderate yield over time in response to altered node number. They showed that increasing the number of nodes beyond 36 per vine (evaluated 24, 36, 48, 60, and 72 retained nodes per vine) for Sauvignon Blanc did not increase yield after the first year. This concluded that cane vigor and fruit clusters per vine are part of the mechanisms by which vines moderate yield over time in response to altered node number. Even though a longer study would have been beneficial for the determination of sustainability for the primary bud retention levels, the lack of a yield decrease the second year may suggest that the primary bud retention levels did not reach an over-cropping level, especially with the warmer growing conditions in 2012. Higher crop load levels would have helped to delineate over-cropping, but with severe winter dieback observed for Marquette vines in a cultivar trial the winter of 2009/2010 [30], and the desire to not cause dieback to vines owned by growers in the area, selected primary bud retention levels may have been too similar to adequately define optimal crop load for the two cultivars.

Fruit-zone leaf removal influence on yield was not expected, as research with *V. vinifera* cultivars showed yield reductions only when fruit-zone leaves were removed pre-flowering or at flowering [37,38]. Kosteridis et al. [38] did show that post-flowering leaf removal decreased yield per vine and cluster weight in Merlot and Sangiovese. Scharfetter et al. [19] reported numerical increases in yield per vine with fruit-zone leaf removal for several cold-climate interspecific hybrid cultivars, but the variability over the three years eliminated any leaf removal differences. None of the research on leaf removal had as short of a growing season or as low accumulated growing degree-days (AGGD) as reported in the current study and little research has examined carbon partitioning in response to fruit-zone leaf removal. Chanishvili et al. [39] reported that the defoliation of a *V. vinifera* cultivar created a powerful sink for assimilates, enhanced the photosynthetic activity of the remaining source leaves and redirected assimilate transport toward the defoliation zone, from the young leaves in particular. This is understandable for grapes grown in warm and hot environmental conditions, but the average monthly temperature for the three locations in July was 22 °C. Motomura [40] showed that the growth of set berries until they reach at least 30%−40% of their final size, primarily relies on carbon assimilated by leaves right above and below the cluster. Frioni et al. [41] showed that in the temperate climate of Michigan a severe level of Pinot Noir leaf removal (10 leaves) at flowering drastically decreased fruit-set and final yield. However, they could not explain why the removal of six basal leaves at flowering (removing 47% of the pending leaf area) had no significant effect on fruit-set or final yield. The lowest yield of Frontenac gris occurred with 100% fruit-zone leaf removal, suggesting that leaf removal may have been too severe for the less vigorous cultivar. Splitting the removal of six fruit-zone leaves into two timings may alleviate the low leaf number to fruit cluster ratio and increase carbohydrate synthesis during an important period of fruit development. However, it is unlikely that an additional leaf removal activity is economically feasible, since most vineyards in North Dakota are less than 2 ha and do not have a mechanical method for leaf removal.

Even though average cluster number differences could be explained simply by differences in bud numbers, average cluster weight was not related to bud retention numbers, suggesting that vines may

have been able to support additional clusters if more buds had been retained. Both Marquette and Frontenac gris cluster mass averages were far below their recognized averages of 89 and 131 g per cluster, respectively, while berry masses were near their recognized average of 1.1 g per berry [28]. Berry mass results were counter intuitive to previous research on carbohydrate reserves and pruning methods when yield limits have been reached [6], reinforcing the earlier suggestion of greater bud retention to reach vine balance. Hatterman-Valenti et al. [30] reported that over the span of seven years, Marquette and Frontenac gris cluster mass averages were 65 and 77 g cluster$^{-1}$, respectively, which were also below the recognized averages obtained by researchers at the University of Minnesota. These results suggest that the fewer GDDs and environmental conditions in North Dakota may limit fruit cluster size and production potential.

A higher SSC for Marquette in 2012 compared to 2011 was expected. Marquette is a late maturing cultivar and with the extended length of the growing season in 2012, more soluble solids were assimilated. Marquette SSC at harvest was near the recognized average harvest level of 25.7% [28]. Riesterer-Loper et al. [4] reported higher juice SSC in 2015, but not 2016, for Marquette vines with three fruit-zone leaves removed compared to no leaf removal. Authors attributed this difference to the delay in fruit harvest in 2015. Fruit was harvested two weeks later in 2015, but the AGGD at harvest was only slightly greater, at 1492 GDD in 2015 compared to 1421 GDD in 2016. Scharfetter et al. [19] reported no differences in Marquette fruit SSC in response to fruit-zone leaf removal and speculated that this was due to the removal of only two to three cluster-shading leaves and that the positive effect of leaf removal on fruit composition at harvest only occurs in cool summers, when minimal thermal requirements for fruit maturation were not met. Similar fruit SSC for Frontenac gris in 2011 and 2012 was expected as Haggerty [28] reported similar fruit SSC at harvest for Frontenac gris, but also suggested this was due to the accumulation of at least 1400 GDD each year. Scharfetter et al. [19] also reported similar fruit SSC for Frontenac with and without fruit-zone leaf removal. Frontenac gris was derived as a sport of Frontenac and has been reported to carry many of the same vine growth and fruit chemistry characteristics as Frontenac [42]. Frontenac gris fruit SSC at harvest was not near the recognized average of 26% at harvest, but it was still within the desired level for most wine styles [13].

Haggerty [28] reported that Frontenac gris and Marquette fruit pH profiles varied substantially among years and that a GDD model may not be useful in predicting acidity in the berry. Lack of fruit pH differences at harvest for either cultivar in the current study may suggest that primary bud retention levels, fruit-zone leaf removal levels and environmental conditions interacted or varied enough between the two years that fruit pH differences were not significant. Scharfetter et al. [19] also reported no differences in Marquette fruit pH in response to a leaf removal treatment, while Riesterer-Loper et al. [4] did not report fruit pH.

The lower TA level for Marquette fruit when 100% fruit-zone leaves were removed was expected and previously reported with V. *vinifera* and cold-hardy cultivars [19,20]. However, Riesterer-Loper et al. [4] reported that fruit-zone sunlight exposure did not affect Marquette TA levels. No study has evaluated the TA level for Frontenac gris in response to cluster sunlight exposure. One would assume that since Frontenac gris has been reported to carry many of the same vine growth and fruit chemistry characteristics as Frontenac [43], and the effect of sunlight exposure to clusters has been variable for Frontenac fruit at harvest [3,15,28], Frontenac gris fruit TA in response to fruit-zone leaf removal would also be variable. Haggerty [28] reported that the highest average TA after 1400 GDD were Frontenac(18.2 g/L), Frontenac gris(16.8 g/L), La Crescent(15.2 g/L), and Marquette(12.3 g/L), and that of the cold-hardy hybrid cultivars, St. Pepin, Maréchal Foch, and Marquette were most like V. *vinifera* cultivars in respect to TA.

These results suggest that for the two *V. riparia-based* wine grape cultivars used in this study, environmental conditions such as a shorter growing season and fewer GDDs in North Dakota may reduce the production potential by limiting cluster size without compromising most fruit quality parameters. Primary bud retention levels above those tested in the current study should be investigated for a longer period of time, in order to understand the impact of crop level on Frontenac gris and

Marquette crop yield and fruit composition during the shorter growing season and fewer GDDs in North Dakota. Results also warrant further research into cultivar adaptiveness to northern Great Plains conditions and sunlight exposure manipulation to grape vines and fruit clusters. With further research, it is anticipated that wine grape cultivars and management practices will be identified to withstand the abiotic stresses in North Dakota and northern Great Plains, while still producing sustainable yields and acceptable fruit quality for commercial wine grape production.

**Author Contributions:** Conceptualization, H.H.-V. and A.A.; Methodology, H.H.-V. and A.A.; Data curation, A.A.; Formal analysis, J.H.; Investigation, A.A.; Supervision, H.H.-V.; Review and Editing, H.H.-V. All authors have read and agreed to the published version of the manuscript.

**Funding:** This research was partially funded by the North Dakota Department of Agriculture Specialty Block Grant.

**Acknowledgments:** The authors would like to thank Collin Auwarter, John Stenger, Grant Mehring, and Andrej Svyantek for their assistance and technical support. The authors would also like to thank the owners of Red Trail Vineyard near Buffalo, ND, Twisted Sisters Vineyard near Clifford, ND, and Dakota Breeze Vineyard near Wahpeton, ND, for allowing the experiments with their grapes.

**Conflicts of Interest:** The authors declare no conflict of interest.

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
