# Peer review of "Utilizing Pruning and Leaf Removal to Optimize Ripening of Vitis riparia-Based ‘Frontenac Gris’ and ‘Marquette’ Wine Grapes in the Northern Great Plains"

_horticulturae, doi:10.3390/horticulturae6010018_

Round 1

Reviewer 1 Report

The research problem is very important for science and especially for producers. The results are probably interesting, but the way they are developed is very bad. No analysis of variance declared. Several random one-factor analyses were performed, and in the description the authors indicate that they performed other analyzes - chapter 3.2. Yield; ... three-way interaction - please indicate the table in which these results are presented? I did not analyze the rest of the results in detail - the results presented are fragmentary. On their basis, conclusions cannot be correctly presented.

I do not understand the data in Table 2. This is probably the SAT indicator which is given in hours, but the authors showed that in 2011 Growing Degree Days (10 ° C) was 1371. The description shows that these were days. Interesting!!!

According to the authors: 'Table 7  Effect of year on soluble solid concentration in grapes at harvest averaged over fruit-zone leaf removal and pruning levels' but there are only averages from the years, and in the 'Table 8. Effect of fruit-zone leaf removal and cultivar on titratable acidity at harvest averaged over pruning levels and years at harvest', they show the effect of one of the factors. Lack of logic.

The section of material and methods is chaotic, it requires a lot of correction.

I put specific comments in the text.

All work requires major changes, it cannot be published in this form.

Author Response

Reviewer suggested edits for lines 41, 68, 79, 81, 86, 89, 94, 105, 109, 110, 111, 134, 135, 166, and table 2. The suggested edits were made when possible. Table 2 was clarified as accumulated growing degree days. Throughout the document pruning level was changed to primary buds retained. When questions were asked, I’ve tried to incorporate information to address the questions.

I have read and made major changes to the entire article, especially from the Materials and Methods section onward.

Reviewer 2 Report

The manuscript submitted for review describes the interaction between pruning level (20, 30, and 40 retained buds), fruit-zone leaf removal (0%, 50%, 100%) and growing year (2011 and 2012) on the phenology, yield and fruit quality of two vitis riparia cultivars (Frontenac Gris and Marquette) grown on three vineyards in North Dakota (Buffalo, Clifford, and Wahpeton).

The author collected an impressive amount of data. Unfortunately, it is not easy to understand the manuscript because data on the interaction between the treatments are not presented, some data are provided in the tables and other in the text, and some data are not shown. The vineyard effect on the data is not discussed; it is not clear why different vineyards were studied, why weather data for each vineyard are presented, but not discussed in relation to grape phenology, yield and quality data for each vineyard. The authors are encouraged to provide all the data, including data for each vineyard (as a supplementary information if necessary), which would allow each reader to evaluate the merits of the findings. The advantage in publishing in open access journals is that there is no limit in the number of data that can be shown to substantiate the findings. Moreover, data on the measurements are very useful for meta-analyses, comparisons, or for building food composition databases.

Lines 2-3: All the treatments led to grape ripening. Thus, the title does not reflect the aim of the study as stated in Lines 64-66; the title should be rewritten. Suggestion: “to optimize ripening of” instead of “to ripen”

Lines 2-3: “Vitis riparia” should be added to the title. Not every reader knows that Frontenac Gris and Marquette belong to the species

Lines 10-35: The paragraph is more a summary of the work than an abstract. An abstract of no more than 200 words should be provided as per the journal’s instructions

Line 25: The authors should consider writing: “greater in 2012 compared to 2011”

Line 26: The authors should consider “lowest” instead of “lower”

Lines 39-40: It is important to know when the new industry started in the region, in order to place the reader in the specific context

Lines 40-41: Some of the differences between Vitis riparia and V. vinifera should be provided

Lines 43-44: The sentence should be rewritten. As it read, it seems that the aim of pruning is to influence carbohydrate storage. Suggestion: remove “because”.

Line 45: some of these areas should be listed with the corresponding references

Line 49: “and more specifically by winter dieback”. This end of the sentence does not reconcile well with the beginning, and can be deleted

Line 52: The expression “sustainable balance” should be defined

Line 53-54: “that make up the majority of the North Dakota growing grape and wine industry”. As it read, it seems the study is directed at North Dakota winegrowers. This end of the sentence can be removed

Line 58: The authors should consider; “increased rate of berry ripening” instead of “increased berry ripening”

Line 60: The authors should consider: “fruit to high solar radiations” instead of “fruit to increased solar radiation”

Lines 69-77: From the description, there are at least two different soil types at each vineyard. It is important to know how the soils were distributed in the same vineyard and it is more relevant to the reader to know the specific soil at the plots/vines used for the study

Lines 79-80: The age of each vineyard should be provided

Line 85: The authors should confirm that all the vineyards had the same vine spacing: 2.44 × 2.44 m

Lines 86-87: The authors should confirm that the two cultivars studied were all grown in all the three vineyards. The distribution of each cultivar in each vineyard should be provided

Lines 90-91: It is important to provide data for each vineyard separately in the “Results” section, and to study the vineyard effect. By providing data for each vineyard and studying the vineyard effect, it would be possible to speculate on the effect of fertilization on the treatments

Line 92: It is important to specify how long the pruning was delayed in 2012 compared to 2011

Lines 101-105: It is important to provide data for each vineyard separately in the “Results” section, and to study the vineyard effect. By providing data for each vineyard and studying the vineyard effect, it would be possible to speculate on the effect of weeding on the treatments

Lines 111-112: Data for light intensity measurements should be provided in the “Results” section

Lines 113-114: The number of reading times/days should be provided

Line 122: Suggestion: “Time of final berry harvest” instead of “Time of harvest”

Line 130: Here, 50 berries are used for TSS and pH measurements instead of 8 berries as in Line 116. The reason for different sampling procedures should be provided

Line 134: The distribution of the blocks in the vineyard should be provided (very important in the experimental design)

Line 136-137: The authors should explain how they proceeded when vine growth/vigor was limited or inconsistent

Line 137-138:” Vineyard” is also an effect that should be discussed as was the “year” effect

Line 143: Suggestion: “between 2011 and 2012” instead of “in 2011 and 2012”

Table 2: The calculation method for GDD should be provided in the “Material and Method” section

Table 3: Rainfall data for some months is missing and the reason should be given

Line 154: “fruit-zone leaf removal level” should be deleted since its effect on the weight of one-year-old canes was not studied

Line 155: Data on “measurement of the longest one-year-old cane on each cordon” (see lines 93-94) should be provided. Such data are very useful for meta-analyses, comparisons, or for building food composition databases. This is clearly highlighted by the authors in Lines 238-242

Table 4: The title should be “Influence of bud retention on weight of pruned dormant canes for two vitis riparia cultivars”. In the table, data for each year (see line 158) and each vineyard should be provided as well as statistical data for the three- or four-ways interaction between all factors (see lines 155-157)

Line 159: “increased training and pruning in 2011”. The sentence is not clear and should be rewritten or explained

Table 5: The title should be “Effect of fruit-zone leaf removal level on fruit yield of two cultivars averaged over pruning levels and years”. Although averaging over pruning levels and years allows to grasp a whole picture of the responses, data for each pruning levels (see lines 176-180), each year and each vineyard should be provided, as well as statistical data for the three- or four-ways interaction between all factors (see lines 165-168)

Line 183: The number of clusters per vines, as well as berry weight data should be provided. Such data are very useful for meta-analyses, comparisons, or for building food composition databases. This is clearly highlighted by the authors in Lines 233-235

Table 6: Data for each pruning levels (see lines 189-191), each year and each vineyard should be provided, as well as statistical data of the three- or four-ways interaction between all factors (see lines 183-184)

Lines 197-198: pH data should be provided. Such data are very useful for meta-analyses, comparisons, or for building food composition databases. This is clearly highlighted by the authors in Lines 245-246

Table 7. “Yield” is not the main factor being studied (see title). The table should be about the effects of fruit-zone leaf removal and bud retention. And of course, data for each year and each vineyard should be provided, as well as statistical data of the three- or four-ways interaction between all factors (see lines 194-195)

Table 7: TSS and pH data collected weekly 21 days post-veraison (see lines 115-118) should be provided and discussed

Lines 202-204: The information is already available in Table 8; the sentence can be deleted

Table 8: Data for each pruning levels (see lines 199-200), each year and each vineyard should be provided, as well as statistical data of the three- or four-ways interaction between all factors (see lines 200-201)

Lines 212-113: The meaning of “lack of differences” should be provided

Lines 218-219: It doesn’t seem that the Ravaz Index was used in the study. The sentence should be rewritten

Line 232: “average cluster weight was not related to bud retention numbers”. The statement seems in contradiction to data in Lines 189-191. How was the relationship assessed?

Lines 233-235: A reference is needed here

Lines 254-258: Results for Frontenac Gris obtained in the study should be discussed with previous results.

Author Response

I've added a file that addresses each of the suggestions by this reviewer. I appreciate the thoroughness of the review and believe the many suggestions really improved the manuscript.

Reviewer 3 Report

Review of Horticulturae-669997

 Utilizing Pruning and Leaf Removal to Ripen  ‘Frontenac Gris’ and ‘Marquette’ Wine Grapes in the  Northern Great Plains

Overall this study is okay and worthy of publication. Unfortunately, it was only done two years and that’s not long enough to really see the effects of pruning severity on vine yield and pruning weights. That said, I think the paper needs major revision before publication.

First, the abstract needs to be re-written. Here’s my suggestion:

Experiments were conducted to evaluate the effects of three pruning levels (20, 30 and 40 nodes per vine) and three fruit-zone leaf removal levels (0, 50, and 100 %) on the yield and fruit quality of ‘Frontenac gris’ and ‘Marquette’ wine grapes in a northern production region. The study was conducted at three North Dakota vineyards located near Buffalo, Clifford, and Wahpeton, North Dakota, in 2011 and 2012. Increasing the number of buds retained increased yields and reduced pruning weights in both cultivars.

‘Frontenac gris’ and ‘Marquette’ yields were greatest when vines had 50% of the fruit-zone leaves removed due to heavier clusters, suggesting that the 100% fruit-zone leaf removal level was too severe. Individual berries in clusters were also heavier when vines were pruned to retain 40 buds. ‘Frontenac gris’ fruit quality was similar both years and was not influenced by pruning or leaf removal levels. ‘Marquette fruit total soluble solids content was greater in 2012 due to the warmer and longer growing season. ‘Marquette’ fruit titratable acidity was lower when 100% of the fruit-zone leaves were removed. These results suggest that for the two cold-hardy hybrid wine grapes used in this study, greater bud retention levels should be investigated. Results also warrant further research into cultivar adaptiveness to northern Great Plains conditions. With further research, it is anticipated that wine grape cultivars and management practices will be identified to produce acceptable yields and fruit quality for commercial wine grape production.

The Introduction is well written and referenced.

Materials and Methods needs major revision. Complete descriptions of the site soils is probably not necessary. I would like to see a discussion of the treatments and plot set up early in this section. What are the treatments? How many vines per plot, how many reps, etc.? Were the treatments combined. E.g. did all node retention treatment have all three leaf removal treatments? That is not clear from the manuscript. And location is not included in any of the analysis results so it is not clear how this was analyzed.

Results:

Weather details are not important and do not need to be included (Tables 1, 2 & 3). A brief description in the text will suffice.

Instead of that data, I want to see tables of the effects of bud retention treatment on yield and pruning weight. This is the most important information from the study. Then we need a table of the effects of leaf removal on yield, cluster wt, berry wt, and fruit quality. Again, that’s why the experiment was done, to see if leaf removal improved fruit quality.

So Table 4 could be bud retention on yield per vine and kg/ha as well as dormant pruning weights on each variety. Then show the interaction significance at the bottom.

Table 5 could be the effects of leaf removal on vine yield, cluster wt, berry wt, TSS, TA and pH.

Then a brief discussion pointing out the main findings.  Too much of the tabled data is reiterated in the text, which is unnecessary.

Author Response

I appreciate the comments of this reviewer and have included my remarks in the attached file.

Reviewer 4 Report

See the file.

Author Response

I have addressed the comments and suggestions on a separate file.

Round 2

Reviewer 1 Report

Dear Autors
I re-read the manuscript and found that my suggestions were taken up in the review.
Best regards

Reviewer 2 Report

Dear Editor,

The revised version of manuscript horticulturae-669997 provided by the author was a more delightful read compared to the previous one. Well, all the suggestions given were taken into consideration. The authors are commended for the effort taken to improve the whole manuscript, and for providing a new “Discussion” section with important clarifications.

The manuscript can be accepted in the present form after the following editorial correction:

Lines 215-216: “Marquette’ vines with 100% fruit-zone leaves removed and 40 buds retained had the heaviest fruit clusters”: It should be 20 buds INSTEAD of 40 buds.

Reviewer 3 Report

Good study and good paper.